# The Middle Eastern Cousin: Comparative Venomics of *Daboia palaestinae* and *Daboia russelii*

**DOI:** 10.3390/toxins14110725

**Published:** 2022-10-23

**Authors:** R. R. Senji Laxme, Suyog Khochare, Saurabh Attarde, Navneet Kaur, Priyanka Jaikumar, Naeem Yusuf Shaikh, Reuven Aharoni, Naftali Primor, Dror Hawlena, Yehu Moran, Kartik Sunagar

**Affiliations:** 1Evolutionary Venomics Lab, Centre for Ecological Sciences, Indian Institute of Science, Bangalore 560012, India; 2Department of Ecology, Evolution and Behavior, Alexander Silberman Institute of Life Sciences, The Hebrew University of Jerusalem, Jerusalem 9190401, Israel; 3Shulov Institute of Science, 10 Oppenheimer Street, Science Park, Rehovot 7670110, Israel

**Keywords:** *Daboia palaestinae*, *Daboia russelii*, snake venoms, venomics, transcriptomes, preclinical assessment

## Abstract

Among the medically most important snakes in the world, the species belonging to the genus *Daboia* have been attributed to the highest number of human envenomings, deaths and disabilities. Given their significant clinical relevance, the venoms of Russell’s vipers (*D. russelii* and *D. siamensis*) have been the primary focus of research. In contrast, the composition, activity, ecology and evolution of venom of its congener, the Palestine viper (*D. palaestinae*), have remained largely understudied. Therefore, to unravel the factors responsible for the enhanced medical relevance of *D. russelii* in comparison to *D. palaestinae*, we comparatively evaluated their venom proteomes, biochemical activities, and mortality and morbidity inflicting potentials. Furthermore, the synthesis and regulation of venom in snakes have also remained underinvestigated, and the relative contribution of each venom gland remains unclear. We address this knowledge gap by sequencing the tissue transcriptomes of both venom glands of *D. palaestinae*, and comparatively evaluating their contribution to the secreted venom concoction. Our findings highlight the disparity in the venom composition, function and toxicities of the two *Daboia* species. We also show that toxin production is not partitioned between the two venom glands of *D. palaestinae*.

## 1. Introduction

The genus *Daboia* is currently constituted by four snake species that are distributed across Asia, the Middle East and Africa: Russell’s viper (*D. russelii*), Siamese or the eastern Russell’s viper (*D. siamensis*), Palestine viper (*D. palaestinae*), and the Moorish viper (*D. mauritanica*). Among them, *D. russelii* is arguably the medically most important species, being responsible for the largest number of human envenomings, deaths and disabilities globally [1]. Considering their relatively greater medical relevance, the two species of Russell’s vipers, *D. russelii* and *D. siamensis*, have been extensively investigated [2,3,4,5,6]. In contrast, the composition, activity, ecology, and evolution of *D. mauritanica* and *D. palaestinae* venom have remained largely uninvestigated. This is despite the fact that the latter is the most common venomous snake species in Israel that causes hundreds of envenomations in humans and livestock. It has also been reported that *D. palaestinae* is responsible for a significant number of snakebites in Lebanon, northwestern Jordan and the Palestinian territories [7,8,9]. Moreover, understanding the mechanisms involving venom production and regulation in snakes has remained the least studied. For instance, despite the presence of a pair of venom glands in snakes, their relative contribution to the secreted venom cocktail remains unclear.

To address this knowledge gap and unravel the influence of phylogenetic divergence and biogeography in shaping the composition and activity of *Daboia* venoms, we comparatively investigated the venoms of *D. russelii* from western India and *D. palaestinae* from Israel, whose range distributions are separated by over 5500 km. Comparative proteomics and biochemical assessment provided fascinating insights into the venom constitution and function. Toxicity profiling and the preclinical evaluation of venom-induced morbidity revealed the underlying factors responsible for the enhanced medical importance of *D. russelii* venoms in the Indian subcontinent. Furthermore, comparative transcriptomics of both venom glands of *D. palaestinae* provided insights into the production and regulation of venom toxins in this species.

## 2. Results

### 2.1. Venom Proteomics

To determine the differences in the proteomic composition of *D. russelii* and *D. palaestinae* venoms, we subjected them to sodium dodecyl sulphate–polyacrylamide gel electrophoresis (SDS-PAGE) and Reversed-phase high-performance liquid chromatography (RP-HPLC). Despite their close phylogenetic relationship, the venom composition of the two *Daboia* species was very distinct (Figure 1A). SDS-PAGE and RP-HPLC profiles of *D. russelii* and *D. palaestinae* venoms unveiled significant differences in patterns and intensities of protein bands (between 10–20 kDa and 50–75 kDa) (Figure 1A; Appendix A) and RP-HPLC peaks (between the retention time of 40 to 80 min) (Figure 1B,C; Appendix A), highlighting the considerable differences in their venom compositions. Moreover, searching the mass spectrometric data from individual gel-excised bands of *D. palaestinae* venom against NCBI-NR Serpentes databases (taxid: 8570) and *D. palaestinae* venom gland transcriptome identified 91 non-redundant protein families (Figure 2A; Appendix A; Appendix A). Among these were 16 toxin families, including cysteine-rich secretory proteins (CRISP), disintegrin, phospholipase A_2_ (PLA_2_), cystatin, vascular endothelial growth factor (VEGF), L-amino acid oxidase (LAAO), nerve growth factor (NGF), Kunitz-type serine protease inhibitor (Kunitz), 5′-nucleotidase (5′-NT), lectin, hyaluronidase (HYL), phosphodiesterase (PDE), phospholipase B (PLB), serpin, snake venom serine protease (SVSP), and snake venom metalloproteinase (SVMP) (Figure 2A; Appendix A).

Tandem mass spectrometry further revealed that *D. palaestinae* venom is predominantly constituted by SVSP (41%), lectins (29%) and PLA_2_s (12.2%; Figure 2A; Appendix A). Additionally, CRISP, VEGF, SVMP, LAAO, 5′-NT, PDE, NGF, HYL, Kunitz, PLB, serpin, cystatin and disintegrin were also identified as the minor components of the venom proteome (Figure 2A; Appendix A).

### 2.2. Venom Gland Transcriptomics

Tissue transcriptomes of both venom glands from a male *D. palaestinae* were sequenced on an Illumina HiSeq 2500 platform. A total of 41,875,666 and 43,278,432 sequences were retrieved from the left (VG1) and right (VG2) venom glands, respectively (Appendix A Appendix A). A combined *de novo* transcriptome from both glands was assembled using Trinity v2.11.0 [10] and a total of 191,425 transcripts were identified. The assembly was characterised by an N50 statistics of 1812 based on all transcripts. The annotation of the resultant venom gland transcripts identified multiple toxin-encoding genes from both glands (Appendix A). *D. palaestinae* venom gland transcriptome profile (VG1 and VG2, respectively) showed the abundance of SVMP (23.5% and 24.3%), lectin (26.2% and 24.4%), SVSP (19.0% and 21.4%) and PLA_2_ (11.7% and 10.4%) transcripts (Figure 2B, Appendix A). Transcripts encoding LAAO, disintegrin, VEGF, Kunitz, PLB, hyaluronidase, PDE, 5′-NT, natriuretic peptide, Kazal-type serine protease inhibitors, CRISPs, cobra venom factor (CVF), waprin and NGF were also retrieved from these glands. Interestingly, we recovered a large number of transcripts encoding snake venom metalloproteinase inhibitors (SVMPI) from the *D. palaestinae* venom glands. Although the role of SVMPIs in venom remains poorly understood, the active-site tripeptide region of a synthetic SVMPI identified from *D. siamensis* has been shown to inhibit SVMPs under experimental conditions. Therefore, SVMPIs may putatively be involved in the inhibition of SVMPs in the venom glands to prevent autotoxicity [11]. SVMPI transcripts are also known to code the precursors of Bradykinin-potentiating peptides (BPP) and natriuretic peptides (NP) in Viperidae snakes [12,13]. Manual inspection of these transcripts in VG1 and VG2, respectively, revealed that most of them encode for SVMPI and BPP (62.49% and 62.49%), followed by SVMPI, BPP and NP (24.99% and 24.99%) and SVMPI only (12.50% and 12.50%).

### 2.3. Venom Biochemistry

#### 2.3.1. PLA_2_ assay

Phospholipases are among the major classes of snake venom toxins that exhibit numerous pharmacological effects, including cytotoxicity, neurotoxicity, myotoxicity and the perturbation of haemostasis. Viperidae snakes are known to possess both catalytic and non-catalytic forms of PLA_2_s [14]. However, in our PLA_2_ assay, both *D. russelii* and *D. palaestinae* venoms exhibited very low PLA_2_ activities (18.27 nmoles/mg/min to 55.27 nmoles/mg/min, respectively; *p* = 0.0058; Figure 3A).

#### 2.3.2. LAAO Assay

LAAOs are flavoproteins that catalyse the stereospecific oxidative deamination of L-amino acids to α-keto acids. Snake venom LAAOs exhibit diverse pharmacological effects, such as oedema, haemorrhage, myotoxicity, apoptosis and necrosis [15,16,17]. In LAAO assays, both *Daboia* venoms showed notable L-amino acid oxidation, albeit the difference between the two was not statistically significant (*p* = 0.6581, Figure 3B).

#### 2.3.3. Snake Venom Protease Assay

Snake venom proteases, such as SVSPs and SVMPs, are known to significantly contribute to the clinical manifestations of snakebite victims [18]. Viperid venoms, in particular, are enriched with these proteolytic enzymes. As a result, they inflict fibrinolysis, inhibition of platelet aggregation, and degradation of the capillary basement membrane [19,20]. When the proteolytic potentials of *D. russelii* and *D. palaestinae* venoms were evaluated in comparison to a bovine pancreatic protease (positive control), both venoms exhibited significant proteolysis, with the *D. russelii* venom exhibiting relatively increased activity than *D. palaestinae* (70% vs. 38%, respectively; *p* < 0.001; Figure 3C). We further assessed the relative contributions of SVMP and SVSP towards proteolysis using ethylenediamine tetraacetic acid (EDTA) and phenylmethylsulfonyl fluoride (PMSF), a metal chelator and protease inhibitor which are known to inhibit these toxins, respectively. While the addition of EDTA completely inhibited the proteolysis of azocasein by *D. russelii* venom, PMSF did not seem to diminish this effect (Figure 3C). Interestingly, both PMSF and EDTA were documented to inhibit proteolytic effects of *D. palaestinae* venom to a similar extent. When both PMSF and EDTA were added to these reaction mixtures, the proteolytic activities of *Daboia* venoms were completely suppressed.

#### 2.3.4. DNase Assay

ETosis is a unique mechanism wherein cells, such as neutrophils, monocytes, or macrophages, release their DNA and granular contents to restrict the movement of venom toxins in the bloodstream. As toxins are trapped at a particular site, the rate of local tissue damage accelerates significantly [21]. Venoms that show DNase activity cause the cleavage of these extracellular traps, leading to the diffusion of venom toxins. Interestingly, certain populations of *D. russelii* in India were found to cleave DNA [22]. However, consistent with the absence of DNase activity in the majority of *Daboia* populations [22], the venoms of *D. russelii* (Maharashtra) and *D. palaestinae* did not exhibit this activity (Appendix A).

#### 2.3.5. Fibrinogenolytic Assay

Fibrinogen, a glycoprotein that forms a hexameric complex (Aα/Bβ/γ)_2_, is cleaved by thrombin when released into the blood. This reaction results in the formation of fibrin monomers and a mesh-like network that ultimately leads to blood clots [23]. Snake venoms can prolong clotting time by cleaving fibrinogen in blood [5,24]. Therefore, we assessed the abilities of *D. russelii* and *D. palaestinae* venoms in cleaving the human fibrinogen. Compared to the control–human fibrinogen with three distinct bands–*D. palaestinae* venom was found to cleave Aα and Bβ components (Appendix A), whereas the *D. russelii* venom exhibited a partial cleavage of Aα subunit retaining the other two components (verified with densitometric analyses of bands; Appendix A). Moreover, when *D. russelii* venom was incubated with EDTA (SVMP inhibitor), all three bands were observed. While the treatment with PMSF (SVSP inhibitor) did not completely inhibit the cleavage of Aα band, the addition of both inhibitors showed all three bands, suggesting the complete inhibition of fibrinogenolytic activity (Appendix A).

### 2.4. Preclinical Assessments

#### 2.4.1. Venom Toxicity

Toxicity profiles of *Daboia* venoms were evaluated in the mouse model of envenoming. In these experiments, the venom collected from the western Indian population of *D. russelii* was found to be relatively more toxic (0.20 mg/kg) than the venom of *D. palaestinae* (0.34 mg/kg) from Israel (Figure 4A). Previous studies have highlighted the remarkable biogeographic variation in the lethal potency of *D. russelii* venoms across the Indian subcontinent, ranging from 0.1 mg/kg to 0.4 mg/kg [6,22,25]. Interestingly, toxicities of *D. russelii* and *D. palaestinae* venoms were very similar to those of *D. siamensis* (0.3–0.6 mg/kg) and *D. mauritanica* (0.33 mg/kg) [2,26].

#### 2.4.2. Venom-Induced Morbidity

Envenomations caused due to vipers are also known to inflict morbid symptoms [27]. For instance, over 50% of *Daboia* bite victims are documented to suffer from various forms of morbidities [28]. Considering this, we performed preclinical experiments using *D. russelii* and *D. palaestinae* venoms. Haemorrhagic abilities were observed for both venoms under investigation, and the diameter of lesions was directly proportional to the increasing venom concentrations (Figure 4B,C, Appendix A). The Russell’s viper venom from western India (MHD of 0.48 μg/mouse) was thrice as haemorrhagic as its conger from the Middle East (MHD of 1.48 μg/mouse; Figure 4B,C). Preliminary dose-finding investigations using a single mouse per venom dose did not identify necrotising activities for the *D. palaestinae* venom, even up to 35 μg. Hence, given animal ethics, complete experiments involving five mice per venom dose were not carried out for this species. In contrast, between 20 to 25 μg of the *D. russelii* venom produced significant necrotic lesions in mice (5 mm diameter; Figure 4D; Appendix A).

#### 2.4.3. Venom Induced Nephrotoxicity

As Russell’s vipers from certain regions in India and Israel have been documented to inflict kidney injury in human snakebite victims and livestock [29,30,31], we evaluated the nephrotoxic potentials of *Daboia* venoms using the mouse model. Microscopic examinations of kidney sections of venom-injected mice showed significant pathologies relative to those of the control mice which received normal saline alone. While there was no evidence of inflammation documented in kidneys of the control group, and the morphologies of the renal tubules and glomeruli were also normal (Figure 5A,D), the administration of 10 μg of *D. russelii* (Figure 5B,E) and *D. palaestinae* (Figure 5C,F) venom in the treatment group resulted in shrinkage of the bowman’s capsular space and the formation of haemorrhagic casts. Additionally, changes were also observed in the proximal tubule of the treatment group, such as the loss of proximal brush borders, cytoplasmic vacuolation and formation of interlumen haemorrhagic casts.

## 3. Discussion

### 3.1. Disparate Venoms of D. russelii and D. palaestinae

Despite the vast spatial and biogeographic separation, the venoms of the *Daboia* species across the world share several medically important toxins, such as SVMP, SVSP, lectins, PLA_2_s and Kunitz [3,6,22,32,33,34]. However, the relative abundance of these toxins has been documented to vary significantly between the western Mediterranean and the eastern tropical lineages. Previously, it has been theorised that the venoms of western Mediterranean *Daboia* species, including *D. mauritanica* and *D. palaestinae,* are enriched with SVMPs and lectins, while venoms of eastern tropical *Daboia* snakes are rich in PLA_2_s [32,35]. In line with this hypothesis, we recovered a greater number of SVMP and lectin transcripts from the venom glands of *D. palaestinae* (Figure 2B; Appendix A). However, as evidenced by SDS-PAGE and mass spectrometry profiles, the venom proteome of *D. palaestinae* was found to be rich in SVSPs, lectins and PLA_2_s, rather than SVMPs (Figure 1A and Figure 2A; Appendix A). Such a discrepancy between the snake venom gland transcriptome and proteome has also been previously reported [13,22,36,37,38]. However, in the case of *D. russelii*, varying degrees of SVMPs, ranging from 3 to 22%, have also been recorded across the Indian subcontinent [39,40]. Intraspecific differences in relative toxin constituents have been attributed to a range of ecological and environmental factors [22].

Venoms of *D. palaestinae* and *D. russelii* also varied in their biochemical profiles. For example, the venom of *D. palaestinae* exhibited relatively higher PLA_2_ activity compared to *D. russelii*, whereas the latter species was found to exhibit significantly higher SVMP-mediated proteolytic activity than the former (Figure 3). The prominent role of *D. russelii* SVMPs in inflicting proteolysis was evidenced by the addition of EDTA [41,42]. While EDTA completely inhibited proteolysis of the azocasein substrate, PMSF did not diminish this activity, suggesting that SVMPs are responsible for the proteolytic effects of *D. russelii* venom. Interestingly, the addition of either EDTA or PMSF reduced the overall proteolytic activity of *D. palaestinae* venom to a similar extent, suggesting a synergy between these toxin types. Consistently, when both EDTA and PMSF were added to the reaction, the proteolytic activity of *D. palaestinae* venom was completely suppressed (Figure 3C). Moreover, the fibrinogenolytic effect of *D. palaestinae* venom was inhibited by a combination of EDTA and PMSF, whereas EDTA alone completely inhibited the fibrinogenolytic activity of *D. russelii* venom (Appendix A). It should be noted that given the significant differences documented in the venoms of the pan-Indian populations of *D. russelii* [6,22,25,43], it is likely that the profiles described here do not fully represent the profiles of their species, and variations in the abundance and functions of toxins are expected in accordance with the literature.

### 3.2. Venom Production in D. palaestinae Is Not Partitioned between the Venom Glands

Venom, a complex biochemical cocktail of proteins, carbohydrates, salts and amino acids, is a metabolically taxing trait. To capitalise on the evolutionary advantage of this unique molecular innovation, venomous animals have optimised pathways associated with venom production, storage and delivery. Previous reports have highlighted ingenious adaptations of certain venomous organisms to overcome constraints associated with venom storage and target-specific deployment. Sea anemones and other cnidarians have evolved phyletically unique cells called cnidocytes, or stinging cells, for resource specialisation and partitioning of venom storage [44,45]. Research on the starlet sea anemone, *Nematostella vectensis*, revealed that the early stages of developing cnidocytes exhibit high levels of transcription and translation of venom and structural protein-coding genes [44]. However, since the mature cells have space constraints, given a large centralised capsule, they are marked by decreased levels of both these processes [44]. The geographer cone snail (*Conus geographus*) provides an example of the spatial distinction in venom production to ensure target-specific deployment of venom. The distal region of the venom duct is shown to produce a cocktail exclusively deployed for defensive purposes, while the proximal end is responsible for producing predation-specific venoms [46]. Similarly, in the assassin bug, *Pristhesancus plagipennis*, three unique venom gland lumens have been documented to produce distinct venom cocktails [47]. The venom produced in the posterior main gland has been shown to be responsible for inducing prey paralysis, while the anterior main gland produces defensive venoms [47].

In contrast to the aforementioned venomous animals, advanced snakes with a pair of prominent venom glands, perhaps, do not require partitioning of venom production. To shed light on venom production and resource partitioning, we sequenced transcriptomes of both venom glands of *Daboia palaestinae* (VG1 and VG2, respectively). In line with our hypothesis, a highly similar abundance of venom transcripts across the two venom glands was recovered. The abundance of medically most important toxins, including SVMP (23.5% and 24.3%), lectin (26.2% and 24.4%), PLA_2_ (11.6% and 10.4%), SVSP (19.0% and 21.4%) and Kunitz (6.7% and 6.3%) were comparable across VG1 and VG2 (Figure 2B, Appendix A). Similar chromatography profiles have also been reported for the secretory proteomes of the left and right Duvernoy’s gland from the false coral snake, *Rhinobothryum bovallii* [48]. Similarly, venom fractions collected from the left and right venom glands of *Naja siamensis* exhibited similarities in their composition and receptor-binding activities [49]. A shared venom production strategy between the two glands decreases the metabolic stress on either of the glands while also facilitating the rapid replenishment of their toxin reserve.

### 3.3. The Role of Compositional and Activity Differences in Determining the Clinical Relevance of Daboia Species

Russell’s vipers (*D. russelii* and *D. siamensis*) are amongst snakes capable of delivering the most life-threatening bites to humans. Envenoming by these snakes results in innumerable deaths and immutable morbidities. In India, over 40% of snakebite mortalities have been attributed to *D. russelii* envenoming [50]. Similarly, the western Mediterranean congener of Russell’s viper, the Palestine viper (*D. palaestinae*), is endemic to parts of the Levant and is considered the medically most important snake in Israel. Despite both *D. palaestinae* and *D. russelii* thriving closer to farmlands and human residential habitats, only a few hundred *D. palaestinae* bites are documented in Israel, corresponding to a fraction of *D. russelii* envenoming’s in India [50,51]. This discrepancy in snakebite burden is despite the two species exhibiting a similar toxicity profile against mammals (*D. palaestinae*: 0.34 mg/kg vs. *D. russelii*: 0.20 mg/kg; Figure 4A; Appendix A), and both being capable of injecting nearly 200 mg of the venom in a single bite [22,52].

Envenoming by *Daboia* snakes is typically characterised by local oedema, tissue necrosis, haemorrhage and coagulopathy [6,53,54]. Moreover, these snakes are also known to inflict nephrotoxicity in snakebite victims [29]. In our preclinical assays involving the administration of *Daboia* venoms into mice, both *D. russelii* and *D. palaestinae* venoms were found to cause considerable kidney injury. While the control group that only received normal saline showed no signs of pathology, the administration of minuscule amounts of *Daboia* venom (10 μg/mouse) resulted in significant deformation of the tubular region and glomeruli within 15 min of injection. These alarming findings highlight the severe clinical repercussions of *Daboia* envenoming.

Interestingly, in our preclinical assays, while both *D. russelii* and *D. palaestinae* venoms exhibited haemorrhagic effects, necrotizing activity was only documented in the former species (Figure 4B–D, Appendix A). Clinically, however, *D. russelii* is also known to induce extreme systemic manifestations, such as disseminated intravascular coagulation, capillary leakage syndrome, and intracerebral and subarachnoid haemorrhage [55,56,57]. These differences in the abilities to inflict local and systemic complications could largely explain the relatively greater proportion and severity of morbidities caused by *D. russelii*, in comparison to *D. palaestinae*. Moreover, such differences in venom activities could be dictated by local ecological and environmental factors, including prey availability, predator density, and seasonal and ontogenetic shifts. Consistent with this hypothesis, the amount of venom injected, and the clinical manifestations of *D. palaestinae* bite incidents have been recorded to vary across seasons, with slightly severe symptoms observed in patients bitten during spring or immediately after brumation [8,52].

Moreover, the role of demographics in determining the extent of the human–animal conflict cannot be overstated. With over 1.4 billion people, India is the second-most populous country in the world. It is also home to a vast diversity of snake species that could deliver clinically severe bites to humans. Snakebite is a persistent occupational hazard for farmers and cattle herders of poor agricultural subsistence [58]. Coincidentally, around 60% of the Indian population, who are involved in agriculture and allied activities, reside in rural areas, as opposed to a mere 7% in Israel [59]. The sheer difference in the densities of the human population, and that of the medically relevant snake species in these regions, could explain the significantly disproportionate burden of snakebites in India and Israel. Therefore, in addition to research on snake venom composition, activity and ecology, investigations into the human and snake demography are imperative for the effective management of the prevailing global snakebite crisis.

## 4. Conclusions

In this study, we report a comparative analysis of the venoms of *D. palaestinae* from Israel and *D. russelii* from India, whose distribution ranges are separated by over 5500 km. While the venom of *D. russelii* assessed here was enriched with high molecular weight, haemorrhage-inducing toxins such as SVMPs, the *D. palaestinae* venom was enriched by SVSPs, PLA_2_s and lectins. However, we show that despite the overall similarity in venom functions, the differences in their abilities to inflict local and systemic complications, and the demographics of the vulnerable human populations in the two regions, could explain the disproportionate medical relevance of these snake species. Moreover, this is the first holistic characterisation of the venom of *D. palaestinae*–the most common and medically relevant venomous snake in the State of Israel–using an ‘omics’ approach. Furthermore, comparative tissue transcriptomics revealed the absence of partitioning of venom production between the pair of venom glands in *D. palaestinae*.

## 5. Materials and Methods

### 5.1. Sampling Permits, and the Collection of Venom and Venom Gland

A male Palestine viper (*Daboia palaestinae*) was sourced from the Southern Shfela region of Israel in compliance with the Nature and National Parks Protection Authority (Permit #: 2015/41135). Venom from an adult Russell’s viper (*Daboia russelii*) was collected from the Pune district of Maharashtra in India with prior permission from the state forest department (#Desk-22 (8)/WL/CR-60/(17-18)/2708). The freshly extracted venoms from these individuals were flash-frozen, lyophilised and stored at −80° C until further use. Post milking, *D. palaestinae* was maintained in captivity for three days and humanely euthanised on the fourth day, when the transcriptional activity in the venom glands is believed to be the highest [60]. Euthanisation of the animal was carried out via intramuscular injection of sodium pentobarbital (0.6–0.8 mL/kg) by a licensed veterinarian. After the complete cessation of vital signs and physiological reflexes, both venom glands were dissected, flash-frozen and stored at −80 °C until further processing.

### 5.2. Ethical Statements

The venom toxicity and morbidity assays in mice were performed using the standard protocols recommended by the World Health Organisation (WHO) [61]. The Institutional Animal Ethics Committee (IAEC), Indian Institute of Science (IISc), Bangalore (CAF/Ethics/769/2020; approved on: 16 October 2020), reviewed and approved these protocols. Experiments were also designed adhering to the guidelines issued by the Committee for the Purpose of Control and Supervision of Experiments on Animals (CPCSEA).

### 5.3. Proteomic Analyses

#### 5.3.1. Protein Estimation, One-Dimensional Gel Electrophoresis and In-Gel Digestion

Protein concentrations of venoms were determined using the Bradford method and the Bovine serum albumin (BSA) standard [62]. Variations in the proteomic profiles of *D. palaestinae* and *D. russelii* venoms were assessed using SDS-PAGE. The reduced venom samples (20 µg) were subjected to 12.5% polyacrylamide gel electrophoresis at a constant voltage (80 V) [63]. The Precision Plus Dual Color (Bio-Rad Laboratories, Hercules, CA, USA) protein ladder was used as a reference for determining the molecular weight of proteins. Post-separation, the gels were stained overnight with Coomassie Brilliant Blue R-250 (Sisco Research Laboratories Pvt. Ltd., Mumbai, India) and destained the following day. The protein bands were visualised using an iBright CL1000 gel documentation system (Thermo Fisher Scientific, MA, USA), and densitometric analysis of individual bands was performed using the ImageJ software [64].

The individual protein bands were excised and collected separately for mass spectrometric analyses. Briefly, the gel bands were destained and dehydrated with 50% acetonitrile. Following destaining, proteins were reduced with 10 mM dithiothreitol (DTT) at 56 °C for 1 h and alkylated using 30 mM iodoacetamide (IAA), in the dark at room temperature, for 45 min. Then, the bands were washed with 25 mM ammonium bicarbonate in water and acetonitrile solution (1:1, *v*/*v*), and excess solvent was removed using a vacuum concentrator (Thermo Fisher Scientific, MA, USA). The samples were then digested with trypsin (0.2 µg/µL) overnight at 37 °C, and the peptides were extracted the next day into 50 μL of 50% acetonitrile solution.

#### 5.3.2. Liquid Chromatography-Tandem Mass Spectrometry (LC-MS/MS)

The proteomic composition of individually excised *D. palaestinae* venom bands was characterised using tandem mass spectrometry. Samples (40 µg) were first reduced with 10 mM dithiothreitol (DTT), alkylated using 30 mM iodoacetamide (IAA) and further digested with trypsin (0.2 µg/µL) overnight at 37 °C. The digested samples were run through a C18 nano-LC column (50 cm × 75 µm, 3 µm particle size and 100 Å pore size) with a Thermo EASY nLC 1200 series system (Thermo Fisher Scientific, MA, United States) at a constant flow rate of 300 nl/min for 120 min by varying the concentrations of buffer A (0.1% formic acid in HPLC grade water) and buffer B (0.1% formic acid in 80% acetonitrile) as 10–45% over 98 min, 45–95% over 4 min and 95% over 18 min. Samples were then subjected to tandem mass spectrometry on a Thermo Orbitrap Fusion Mass Spectrometer (Thermo Fisher Scientific, Waltham, MA, USA). The following parameters were defined for the MS scans: range (m/z) of 375–1700 with a resolution of 120,000 and maximum injection time of 50 ms. Furthermore, an ion trap detector with high collision energy fragmentation (30%) was used to perform the fragment scans (MS/MS) with a scan range (m/z) of 100–2000 and a maximum injection time of 35 ms.

The identities of the individual toxins in each of the excised bands were determined by searching the raw MS/MS spectra against the National Center for Biotechnology Information non-redundant (NCBI-NR) Serpentes database (taxid: 8570; with 410,049 entries as of December 2021) and the *D. palaestinae* venom gland transcriptome generated in this study, using PEAKS Studio X Plus (Bioinformatics Solutions Inc., Waterloo, ON, Canada). The following parameters were defined for the search: the parent and fragment mass error tolerance limits were set to 10 ppm and 0.6 Da, respectively. A ‘monoisotopic’ precursor ion search type with ‘semispecific’ trypsin digestion with a maximum of three missed cleavages, cysteine carbamidomethylation (+57.02) as a fixed modification; methionine oxidation (+15.99) as a variable modification was specified. For match acceptance, the filtering parameters were set to a False Discovery Rate (FDR) of 0.1%, detection of ≥1 unique peptide, and a-10lgP protein score of ≥50. The raw mass spectrometry data have been made available at the ProteomeXchange Consortium via the PRIDE partner repository [65] with the data identifier PXD031190 (Reviewer account details: Username: reviewer_pxd031190@ebi.ac.uk; Password: YtCME8qx). Hits with at least one unique matching peptide were considered for downstream analyses. The redundant protein hits from each protein family were removed manually. The relative abundance of each toxin hit in a fraction was determined by estimating the area under the spectral curve (AUC). These AUC values, which represent mean peak intensities, were obtained from PEAKS Studio analyses, and were then normalised across the gel bands using the densitometric estimates from the SDS-PAGE profiles [66]. The relative abundance of a protein family hit (X) was estimated using the equation below, where ‘N’ indicates the number of bands in the SDS-PAGE profile.
$$Relative\;abundance\;of\;X\;\left(\%\right)=\overset{N}{\underset{n=1}{\sum }}\frac{AUC\;of\;X\;in\;Band\;B_{n\;}\times Density\;of\;the\;band\;B_{n}\;\left(\%\right)}{Total\;AUC\;of\;all\;protein\;families\;in\;the\;band\;n\;B_{n}\;}$$

#### 5.3.3. Reversed-Phase High-Performance Liquid Chromatography (RP-HPLC)

Lyophilised *Daboia* venoms (200 μg) were reconstituted in molecular grade water and loaded onto a 4.6 × 250 mm, C18 (5 μm, 300 Å) reversed-phase column attached to a Shimadzu LC-20AD series HPLC system (Kyoto, Japan). The column was equilibrated with solution A [0.1% trifluoroacetic acid (TFA) in water (*v*/*v*)] and the fractions obtained were eluted at a flow rate of 1 mL/min using the graded concentrations of solution B [0.1% TFA in 100% acetonitrile (*v*/*v*)]: 5% for 5 min, 5–15% for 10 min, 15–45% for 60 min, and 45–70% for 10 min and 70% for 5 min at a flow rate of 1 mL/min. The absorbance was monitored at 215 nm.

### 5.4. Transcriptomics Analyses

#### 5.4.1. RNA Isolation and Sequencing

The total RNA was extracted from freshly preserved venom gland tissues of *D. palaestinae* using the TRIzol™ Reagent (Invitrogen, Thermo Fisher Scientific, Waltham, MA, USA) following the manufacturer’s protocol. Isolated RNA samples were treated with Turbo DNase (Thermo Fisher Scientific, Waltham, MA, USA) to remove DNA contamination, followed by another round of extraction with TRIzol™. The concentration and purity of the isolated RNA were measured using an Epoch 2 microplate spectrophotometer (BioTek Instruments, Inc., Winooski, VT, USA), and the integrity was assessed on a bioanalyzer Agilent 4200 TapeStation system using RNA HS ScreenTape (Cat# 5067-5579; Agilent Technologies, Santa Clara, CA, USA). RNA samples that had an RNA Integrity Number (RIN) of greater than 8 were used to generate cDNA library using the NEBNext^®^ Ultra™ RNA Library Prep Kit (New England Biolabs, Ipswich, MA, USA) for Illumina^®^ and subsequently sequenced on an Illumina HiSeq 2500 platform, with a sequencing depth of 20 million reads (2 × 150 bp paired-end). Raw sequencing results are deposited to NCBI’s Sequence Read Archive (SRA) repository: Bioproject: PRJNA800175; SRA: SRR17903474 and SRR17903475.

#### 5.4.2. Quality Assessment, De Novo Assembly and Annotation

The acquired raw data were screened for high-quality reads using Trimmomatic v0.39 [67]. The filtering process involved the removal of adapter sequences, trimming leading and trailing low-quality bases (<3), and discarding short (<20 nucleotides) and low-quality reads (<25; sliding window of 4). The quality of the processed data was then assessed with FASTQC v0.11.9 [68], before and after trimming. The curated reads were *de novo* assembled into contigs using Trinity v2.11.0 [69] with the following parameters: k-mer = 25, minimum k-mer coverage = 1, minimum contig length = 200, pair distance = 500 and the maximum number of reads per graph = 200,000. Reads were aligned back onto the transcriptome using bowtie v2.4.2 [70] to evaluate the quality of the assembly. TransDecoder v5.5.0 [10] was used to predict the coding regions in transcripts that code for a contiguous stretch of over 30 amino acids. The coding transcripts were annotated with BLAST searches [71] against the SwissProt (December 2021) and NCBI’s non-redundant protein databases.

#### 5.4.3. Quantification and Differential Expression Analyses

The abundance of transcripts in Fragments per kilobase of exon per million fragments mapped (FPKM) was quantified using RSEM v1.3.3 [69]. Venom gland-specific expression was determined by pairwise differential expression analysis performed using the EdgeR package within the Bioconductor tool [72]. A predefined cut-off value for fold change in expression (≥2) and probability of *p* ≥ 0.9 was used to identify significantly differentially expressed transcripts.

### 5.5. Biochemical Characterisation

#### 5.5.1. Colourimetric Phospholipase A_2_ (PLA_2_) Assay

The phospholipase activity of venom PLA_2_ was assessed using a chromogenic lipid substrate, 4-nitro-3-[octanoyloxy] benzoic acid (NOB; Enzo Life Sciences, New York, NY, USA). Briefly, 5 µg of the venom was added to 500 mM NOB substrate dissolved in a 200 µL reaction buffer (10 mM Tris–HCl, 10 mM CaCl_2_, 100 mM NaCl, pH 7.8). The mixture was incubated at 37 °C for 40 min, and the kinetics of the assay was monitored by measuring the absorbance at 425 nm every 10 min using an Epoch 2 microplate spectrophotometer (BioTek Instruments, Inc., Winooski, VT, USA). A blank containing only the chromogenic substrate and the buffer without any enzyme was included in the experiment. The endpoint reading at the 40th min was considered for calculating the specific PLA_2_ activity after subtracting the blank. An identical protocol was followed, and a standard curve with varying concentrations of the NOB substrate (4 nanomoles to 130 nanomoles) and 4M NaOH was plotted. The amount of the phospholipid substrate in nmol cleaved per minute per mg of the venom was calculated by extrapolation from the standard curve [73,74].

#### 5.5.2. L-amino Acid Oxidase (LAAO) Assay

The LAAO activity of *D. russelii* and *D. palaestinae* venoms was evaluated using a previously described endpoint assay [22]. The L-leucine substrate solution (5 mM L-leucine, 50 nM Tris-HCl buffer, 5 IU/mL horseradish peroxidase, 2 mM o-phenylenediamine dihydrochloride) was incubated with 10 microlitres of the crude venom at 37 °C. After one hour of incubation, the reaction was terminated by the addition of 2M H_2_SO_4_ solution. The absorbance of the solution was recorded at 492 nm using an Epoch 2 microplate spectrophotometer (BioTek Instruments, Inc., Winooski, VT, USA).

#### 5.5.3. Snake Venom Protease Assay

The snake venom protease activity was assessed using previously established protocols [75], wherein a known amount of the crude venom (10 µg) was incubated with the azocasein substrate at 37 °C for 90 min. Post-incubation, the reaction was stopped by adding trichloroacetic acid (200 µL). Before the addition of azocasein, both venoms were incubated at 37 °C for 15 min with 0.1 M EDTA and 0.04 M PMSF to assess the contribution of SVMP and SVSP, respectively, towards the overall proteolytic activity [41,42]. This mixture was then subjected to centrifugation at 1000× *g* for 5 min. The supernatant was mixed with equal volumes of 0.5 M NaOH, and the absorbance was measured in an Epoch 2 microplate spectrophotometer (BioTek Instruments, Inc., Winooski, VT, USA) at 440 nm. Purified bovine pancreatic protease (Sigma-Aldrich, Burlington, MA, USA) was used as a positive control, and the relative proteolytic activities of *Daboia* venoms were calculated.

#### 5.5.4. DNase Assay

The DNase assay was conducted on *D. russelii* and *D. palaestinae* venoms using a previously described protocol [76]. A known concentration of the crude venom was added to the purified calf thymus DNA (Sigma-Aldrich, Burlington, MA, USA) dissolved in phosphate buffer saline (PBS; pH 7.4), and the reaction mixture was incubated at 37 °C for 60 min. This mixture was thereafter run on a 0.8% agarose gel electrophoresis and imaged using an iBright CL1000 (Thermo Fisher Scientific, MA, USA). Intact calf thymus DNA, and DNase I from the bovine pancreas (15 U), were used as negative and positive controls, respectively.

#### 5.5.5. Fibrinogenolytic Assay

The fibrinogenolytic activity of *Daboia* venoms was visualised electrophoretically [77]. Briefly, 1.5 µg of crude venom in phosphate buffer saline (PBS, pH 7.4) was incubated with 15 µg of human fibrinogen (Sigma-Aldrich, Burlington, MA, USA) at 37 °C for 60 min. We also preincubated the venom with 0.1 M EDTA (SVMP inhibitor) and/or 0.04 M PMSF (SVSP inhibitor) at 37 °C for 15 min to assess the respective contributions of SVMP and SVSP towards proteolysis [78]. The reaction was stopped by adding an equal volume of sample loading buffer (1 M Tris-HCl, pH 6.8; 50% Glycerol; 0.5% Bromophenol blue; 10% SDS; and 20% β-mercaptoethanol) and heated at 70 °C for 10 min. A 15% polyacrylamide gel was run, and the banding patterns of the fibrinogen cleavage products were observed by comparing them to an untreated human fibrinogen control. Densitometric analyses of the bands were carried out using the ImageJ tool [64].

### 5.6. Preclinical Assessments

#### 5.6.1. The Median lethal Dose (LD_50_)

The median lethal dose or the LD_50_ of the venom, which is defined as the minimum amount of venom that can kill 50% of the test population, was determined using the murine model of envenoming [61]. Five distinct concentrations of *D. russelii* and *D. palaestinae* venoms, prepared in the physiological saline (0.9% NaCl), were administered intravenously into the caudal vein of male CD-1 mice (200 µL/mouse). Death and survival patterns were recorded for each venom dose group (n = 5), 24 h post-venom injection. Finally, using Probit analysis, the LD_50_ values were calculated with 95% confidence intervals [79].

#### 5.6.2. The Minimum Haemorrhagic Dose (MHD)

MHD is defined as the amount of venom in μg that induces a 10 mm haemorrhagic lesion within three hours of intradermal injection in mice [61,80,81]. To determine the MHD of *D. russelii* and *D. palaestinae* venoms, five graded venom concentrations were dissolved in the physiological saline (50 µL) before being intradermally injected into a group of five male mice (CD-1 mice; 18–22 gm). The group, where only the physiological saline was administered, served as the negative control. Three hours post-venom injection, mice were humanely euthanised with CO_2_ asphyxiation, and the diameter of the haemorrhagic lesion on the dorsal skin patch was measured using a vernier calliper.

#### 5.6.3. The Minimum Necrotic Dose (MND)

MND is defined as the amount of venom in μg that induces a 5 mm necrotic lesion within 72 h of intradermal injection in mice [61,80]. To determine the MND of *Daboia* venoms, five graded venom concentrations were dissolved in physiological saline (50 µL) and intradermally injected into a group of five male mice (CD-1 mice; 18–22 gm). The group receiving physiological saline alone served as the negative control. Mice were euthanised humanely, 72 h post-venom injection, and the dorsal skin patch was examined for necrotic lesions.

### 5.7. Nephrotoxic Potentials of Daboia Venoms

The nephrotoxic effects of *D. russelii* and *D. palaestinae* venoms were assessed by injecting 10 μg of the venom into the caudal vein of 4 male CD-1 mice (18–22 g). Both kidneys were harvested immediately after the death of the animal and were washed with 1X PBS, fixed in 10% buffered formalin for 24 h, and dehydrated with ascending concentrations of ethyl alcohol (70 and 95% for 30 min; 100% for 2 h), and cleared in xylene (Thermofisher, Waltham, MA, USA). Tissues were embedded in paraffin (Thermofisher, Waltham, MA, USA) at 58 °C, following which 3 μm sections were prepared using a Leica microtome (RM2245, Wetzlar, Germany). The slides were then stained with hematoxylin (Leica, Wetzlar, Germany), eosin (Leica, Wetzlar, Germany) and Masson’s trichrome staining (MTS; Path Stains, Bengaluru, India). The slides obtained were visualised using an Olympus light microscope (Ix81, Olympus, Shinjuku, Japan) at a 40× magnification, and images were acquired and analysed using CellSens dimension imaging software (Olympus, Shinjuku, Japan). The histological structure of renal tubules and glomeruli of the treatment group (10 μg of venom) was assessed in comparison to the control that received 200 μL of normal saline [82].

### 5.8. Statistical Analysis

Statistical comparisons between samples were carried out using an unpaired *t*-test and one-way ANOVA in GraphPad Prism (GraphPad Software 9.0, San Diego, CA USA, www.graphpad.com, accessed on 22 September 2022).

## Figures and Tables

**Figure 1 toxins-14-00725-f001:**
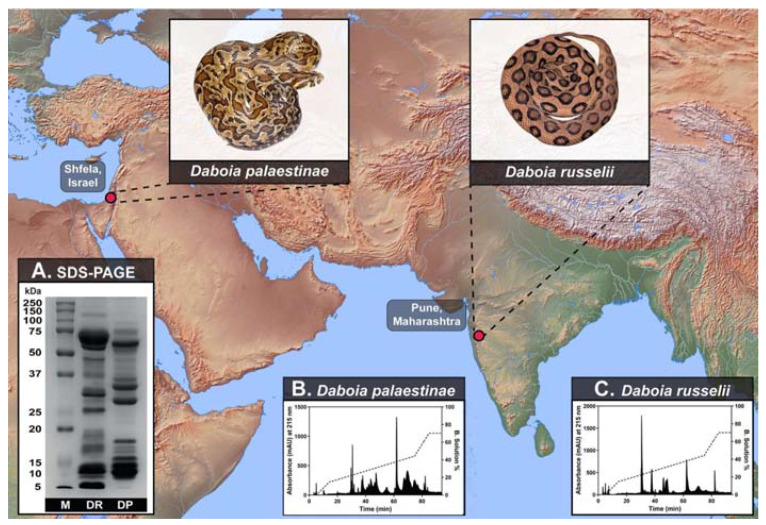
Sampling locations of *D. palaestinae* (Israel) and *D. russelii* (India), along with their representative photographs, are shown. Panel (**A**) depicts the comparative SDS-PAGE profiles of *D. palaestinae* (DP) and *D. russelii* (DR) venoms (M: protein marker), while their respective RP-HPLC profiles are shown in panels (**B**,**C**).

**Figure 2 toxins-14-00725-f002:**
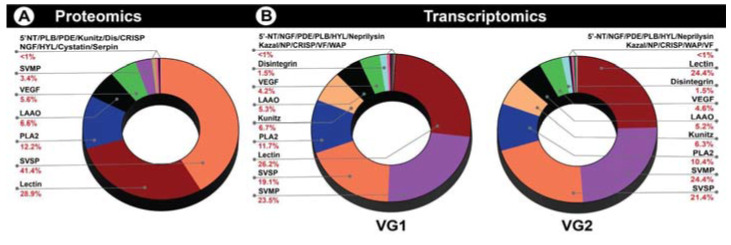
Venom proteome and venom gland transcriptomes of *D. palaestinae.* Here, the doughnut charts indicate the relative abundance of various toxins in the (**A**) venom profile of *D. palaestinae* and (**B**) tissue transcriptomes of the left and right venom glands of this species.

**Figure 3 toxins-14-00725-f003:**
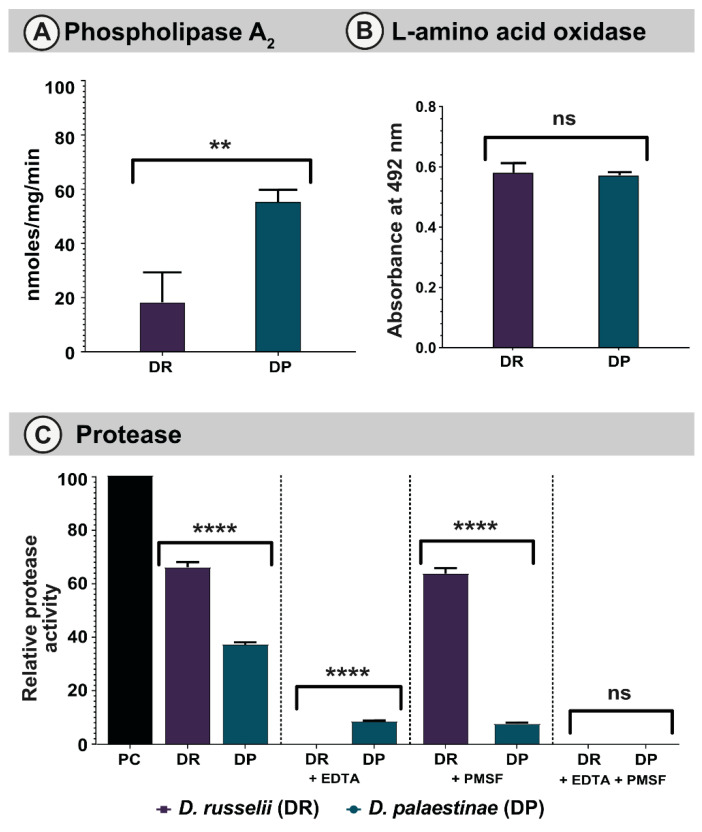
Biochemical activities of *Daboia* venoms. This figure depicts (**A**) PLA_2_, (**B**) LAAO, and (**C**) proteolytic (with and without SVMP and SVSP inhibitors) activities of *Daboia* venoms. Here, the standard deviation is represented as error bars (PC: positive control; DR: *D. russelii*; DP: *D. palaestinae*). The statistical significance is represented as follows: *p* < 0.01 and 0.0001 are indicated as ** and ****, respectively. ns indicates statistical insignificance.

**Figure 4 toxins-14-00725-f004:**
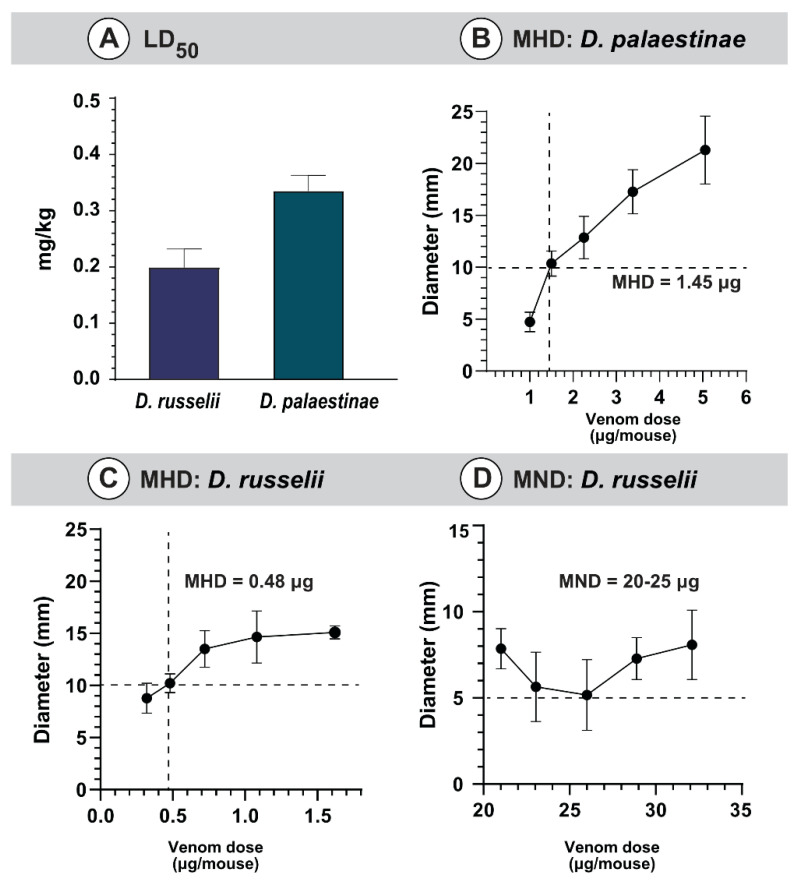
Preclinical assessment of *Daboia* venoms. This figure depicts the (**A**) LD_50_ of *D. russelii* and *D. palaestinae* venoms, MHDs of (**B**) *D. palaestinae* and (**C**) *D. russelii* venom, and the (**D**) MND of *D. russelii* venom.

**Figure 5 toxins-14-00725-f005:**
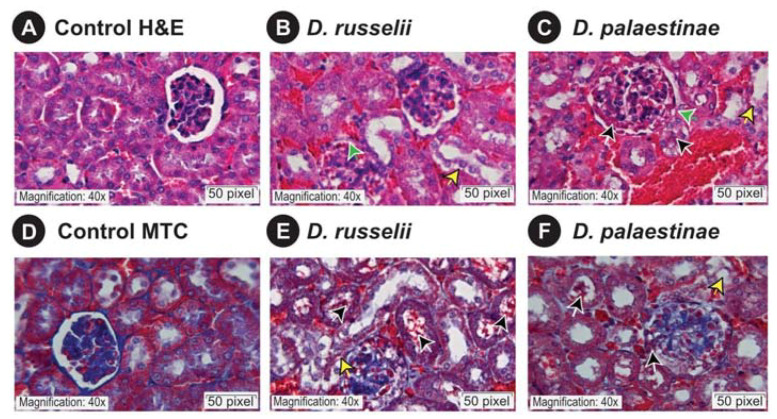
Microscopic observations of mouse kidney sections. This figure shows hematoxylin-eosin and Masson’s trichrome stained kidney sections of the control mice (**A**,**D**), and *D. russelii* (**B**,**E**) and (**D**)*. palaestinae* (**C**,**F**) venom-injected mice. A scale bar of 20 μm is shown, along with green, yellow and black arrows that indicate glomerular capsular space shrinkage, tubular injury and cellular debris, respectively.

## Data Availability

The raw proteomics data generated for this study can be found at PRIDE Database (Accession No: PXD031190). The transcriptomics data presented in this study can be openly accessed via Sequence Read Archive (SRA) at NCBI (Bioproject: PRJNA800175; SRA: SRR17903474 and SRR17903475).

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
