# Peer review of "The Middle Eastern Cousin: Comparative Venomics of Daboia palaestinae and Daboia russelii"

_toxins, 2022, doi:10.3390/toxins14110725_

Round 1

Reviewer 1 Report

The biochemical and toxicological characterization of medically important venoms is a key step in tackling snakebites. This study describes the venom composition at proteomic and transcriptomic levels, with emphasis on the enzymatic and damaging activities induced by crude venoms.

1. The chromatogram does not agree with the run time described in the methodology. The last minutes of elution and the gradient were omitted from the methodology and must be included.

2. Figure 1 emphasizes the regions where the snake venoms were obtained. However, the study only considers a sample of each venom. In this sense, variability is not the main focus of the investigation. On the other hand, the chromatographic profiles are not clear and must be the key part of this figure.

3. The authors present results in the introduction, which makes the manuscript repetitive. The results are already in the abstract, in the results and in the conclusion. Therefore, I suggest removing this part from the introduction.

4. What time was considered for the calculation of phospholipase activity? The authors monitored for 40 minutes, however it is not clear which reading is considered to express this result.

5. The PLA2 activity of the venoms is extremely low in contrast to the expression of the proteomic profile. Samples with the similar proportion of this type of enzyme have higher enzymatic activities. Authors should include data on the proportion of the different isoforms identified in their proteomic study. Are most Lys49 or Asp49 PLA2s? Are there more acidic or basic PLA2s?

6. Some parts of the results include discussion.

7. In figure 7 the authors expressed concentration as ug. However this is not a unit of concentration. Authors should express it as a dose (ug/mice) or consider the injection volume and express the concentration (ug/mL).

8. Some chromogenic substrates degrade after a certain period of incubation even in the absence of enzymes. Did the authors perform negative controls?

9. Statistical analysis must be included in the figures. For example figures 3C and 4A.

Author Response

Please find the attached response.

Reviewer 2 Report

The authors have provided the first characterisation of the venom proteome and transcriptome of the medically significant viper Daboia palaestinae, complete with functional studies and a comparison with its congener D. russelii. The study was scientifically sound and will be of interest to researchers in this field. As minor improvements I would suggest increasing the scale of the Y axis in Figure 3A of PLA2 activity as both values are under 100 nmoles/mg/mL and the Y axis is currently 500 which is excessive. I would also suggest that the authors include more of their conclusions in the abstract which currently only addresses background/methods. Also emphasise the results in more detail in the conclusion.

Author Response

Please find the attached response.

Reviewer 3 Report

Manuskript-ID:   toxins-1960266

Journal:               Toxins

Title:                    The Middle Eastern cousin: Comparative venomics of  Daboia palaestinae and Daboia russelii

Authors:              unknown

Short Summary

The authors submitted a manuscript that describes an "omics"-based investigation of the venom composition of Daboia palaestinae, the Palestine viper. Both proteomic and transcriptomic data were used for the determination of qualitative and quantitative abundances of distinct classes of snake venom toxins. The proteom-based venom composition of Daboia palaestinae was compared with the respective venom composition of Daboia palaestinae, the Indian Russell`s viper. Subsequent functional analyses using in vitro and in vivo assays revealed remarkable differences between the two venoms that may, at least in part, explain the differences between bites of both snake species in terms severity and morbidity. A comparative transcriptome analysis of the two venom glands of D. palaestinae revealed an even contribution by both glands to the final venom composition.

General Impression

The manuscript addresses a topic of current interest that is of relevance for both basic research on one hand and for public health on the other hand. In that context I like the general aim of the manuscript very much and strongly recommend its publication. I do not have any doubts that the authors performed their study with accuracy and used all state-of-the-art techniques appropriately. The manuscript is well written and the rationales behind every experimental step are easy to follow.

There are only a few comments and suggestions that I want to make, most of them are easily to implement.

Comments and minor concerns:

1) The data of the study were obtained from the analyses of one individual for each species. It is well documented and correctly emphasized in the manuscript (e.g. reference 22) that the snake venom composition is influenced by biogeographic parameters and hence may vary between individuals of the same species and between members of different populations. To what extend the results of the current investigation, especially the differences in venom composition, are representative for the respective species?

2) lane 6: the phrase "relatively increased clinical relevance" sounds a little weird

3) lanes 40-41: the sentence "Moreover, venom production and regulation have remained the least studied in snakes." sounds a little weird, too.

4) The values for PLA2 activities are either given in absolute (nmoles/mg/ml, Figure A) or relative (3.6% and 11%, lane 119) values. Please describe how the relative values were derived from the absolute values.

5) lanes 235-237: Is there a mechanistic explanation (e.g. transcription efficiency) for the differences between transcriptome and proteome abundances of snake venom components?

6) lane 314-315: The phrase "only D. russelii venom inflicted haemorrhagic and necrotic effects" is in contrast to the results of chapter 2.4.2 and Figure S5. Please correct.

7) lane 354: "Venom from an adult Russell’s viper" Please specify the sex.

Author Response

Please find the attached response.

Round 2

Reviewer 1 Report

The new version of this manuscript has significantly improved. The authors have carefully addressed most of my concerns.